# Learning Populations of Parameters

**Kevin Tian, Weihao Kong, and Gregory Valiant**
Department of Computer Science
Stanford University
Stanford, CA, 94305
(kjtian, whkong, valiant)@stanford.edu

## Abstract

Consider the following estimation problem: there are $n$ entities, each with an unknown parameter $p_i \in [0, 1]$, and we observe $n$ independent random variables, $X_1, \ldots, X_n$, with $X_i \sim \text{Binomial}(t, p_i)$. How accurately can one recover the "histogram" (i.e. cumulative density function) of the $p_i$'s? While the empirical estimates would recover the histogram to earth mover distance $\Theta(\frac{1}{\sqrt{t}})$ (equivalently, $\ell_1$ distance between the CDFs), we show that, provided $n$ is sufficiently large, we can achieve error $O(\frac{1}{t})$ which is information theoretically optimal. We also extend our results to the multi-dimensional parameter case, capturing settings where each member of the population has multiple associated parameters. Beyond the theoretical results, we demonstrate that the recovery algorithm performs well in practice on a variety of datasets, providing illuminating insights into several domains, including politics, sports analytics, and variation in the gender ratio of offspring.

## 1 Introduction

In many domains, from medical records, to the outcomes of political elections, performance in sports, and a number of biological studies, we have enormous datasets that reflect properties of a large number of entities/individuals. Nevertheless, for many of these datasets, the amount of information that we have about each entity is relatively modest—often too little to accurately infer properties about that entity. In this work, we consider the extent to which we can accurately recover an estimate of the *population* or *set* of property values of the entities, even in the regime in which there is insufficient data to resolve properties of each specific entity.

To give a concrete example, suppose we have a large dataset representing 1M people, that records whether each person had the flu in each of the past 5 years. Suppose each person has some underlying probability of contracting the flu in a given year, with $p_i$ representing the probability that the $i^{th}$ person contracts the flu each year (and assuming independence between years). With 5 years of data, the empirical estimates $\hat{p}_i$ for each person are quite noisy (and the estimates will all be multiples of $\frac{1}{5}$). Despite this, to what extent can we hope to accurately recover the population or set of $p_i$'s? An accurate recovery of this population of parameters might be very useful—is it the case that most people have similar underlying probabilities of contracting the flu, or is there significant variation between people? Additionally, such an estimate of this population could be fruitfully leveraged as a prior in making concrete predictions about individuals' $p_i$'s, as a type of *empirical Bayes* method.

The following example motivates the hope for significantly improving upon the empirical estimates:

**Example 1.** *Consider a set of $n$ biased coins, with the $i^{th}$ coin having an unknown bias $p_i$. Suppose we flip each coin twice (independently), and observe that the number of coins where both flips landed heads is roughly $\frac{n}{4}$, and similarly for the number coins that landed $HT, TH$, and $TT$. We can safely conclude that almost all of the $p_i$'s are almost exactly $\frac{1}{2}$. The reasoning proceeds in two*

*steps: first, since the average outcome is balanced between* $\mathrm{heads}$ *and* $\mathrm{tails}$, *the average* $p_i$ *must be very close to* $\frac{1}{2}$. *Given this, if there was any significant amount of variation in the* $p_i$'s, *one would expect to see significantly more* $HH$s *and* $TT$s *than the* $HT$ *and* $TH$ *outcomes, simply because* $\Pr[Binomial(2,p) = 1] = 2p(1-p)$ *attains a maximum for* $p = 1/2$.

*Furthermore, suppose we now consider the* $i^{th}$ *coin, and see that it landed heads twice. The empirical estimate of* $p_i$ *would be* $1$, *but if we observe close to* $\frac{n}{4}$ *coins with each pair of outcomes, using the above reasoning that argues that almost all of the* $p$'s *are likely close to* $\frac{1}{2}$, *we could safely conclude that* $p_i$ *is likely close to* $\frac{1}{2}$.

This ability to "denoise" the empirical estimate of a parameter based on the observations of a number of *independent* random variables (in this case, the outcomes of the tosses of the other coins), was first pointed out by Charles Stein in the setting of estimating the means of a set of Gaussians and is known as "Stein's phenomenon" [14]. We discuss this further in Section 1.1. Example 1 was chosen to be an extreme illustration of the ability to leverage the large number of entities being studied, $n$, to partially compensate for the small amount of data reflecting each entity (the 2 tosses of each coin, in the above example).

Our main result, stated below, demonstrates that even for worst-case sets of $p$'s, significant "denoising" is possible. While we cannot hope to always accurately recover each $p_i$, we show that we can accurately recover the *set* or *histogram* of the $p$'s, as measured in the $\ell_1$ distance between the cumulative distribution functions, or equivalently, the "earth mover's distance" (also known as 1-Wasserstein distance) between the set of $p$'s regarded as a distribution $P$ that places mass $\frac{1}{n}$ at each $p_i$, and the distribution $Q$ returned by our estimator. Equivalently, our returned distribution $Q$ can also be represented as a set of $n$ values $q_1, \ldots, q_n$, in which case this earth mover's distance is precisely $1/n$ times the $\ell_1$ distance between the vector of sorted $p_i$'s, and the vector of sorted $q_i$'s.

**Theorem 1.** *Consider a set of $n$ probabilities, $p_1, \ldots, p_n$ with $p_i \in [0, 1]$, and suppose we observe the outcome of $t$ independent flips of each coin, namely $X_1, \ldots, X_n$, with $X_i \sim Binomial(t, p_i)$. There is an algorithm that produces a distribution $Q$ supported on $[0, 1]$, such that with probability at least $1 - \delta$ over the randomness of $X_1, \ldots, X_n$,*

$$\|P - Q\|_W \le \frac{\pi}{t} + 3^t \sum_{i=1}^{t} \sqrt{\ln(\frac{2t}{\delta})\frac{3}{n}} \le \frac{\pi}{t} + O_\delta(\frac{3^t t \ln t}{\sqrt{n}}),$$

*where $P$ denotes the distribution that places mass $\frac{1}{n}$ at value $p_i$, and $\|\cdot\|_W$ denotes the Wasserstein distance.*

The above theorem applies to the setting where we hope to recover a set of arbitrary $p_i$'s. In some practical settings, we might think of each $p_i$ as being sampled independently from some underlying distribution $P_{pop}$ over probabilities, and the goal is to recover this population distribution $P_{pop}$. Since the empirical distribution of $n$ draws from a distribution $P_{pop}$ over $[0, 1]$ converges to $P_{pop}$ in Wasserstein distance at a rate of $O(1/\sqrt{n})$, the above theorem immediately yeilds the analogous result in this setting:

**Corollary 1.** *Consider a distribution $P_{pop}$ over $[0, 1]$, and suppose we observe $X_1, \ldots, X_n$ where $X_i$ is obtained by first drawing $p_i$ independently from $P_{pop}$, and then drawing $X_i$ from $Binomial(t, p_i)$. There is an algorithm that will output a distribution $Q$ such that with probability at least $1 - \delta$,* $\|P_{pop} - Q\|_W \le \frac{\pi}{t} + O_\delta\left(\frac{3^t t \ln t}{\sqrt{n}}\right)$.

The inverse linear dependence on $t$ of Theorem 1 and Corollary 1 is information theoretically optimal, and is attained asymptotically for sufficiently large $n$:

**Proposition 1.** *Let $P_{pop}$ denote a distribution over $[0, 1]$, and for positive integers $t$ and $n$, let $X_1, \ldots, X_n$ denote random variables with $X_i$ distributed as $Binomial(t, p_i)$ where $p_i$ is drawn independently according to $P_{pop}$. An estimator $f$ maps $X_1, \ldots, X_n$ to a distribution $f(X_1, \ldots, X_n)$. Then, for every fixed $t$, the following lower bound on the accuracy of any estimator holds for all $n$:*

$$\inf_f \sup_{P_{pop}} \mathbb{E}\left[\|f(X_1, \ldots, X_n) - P_{pop}\|_W\right] > \frac{1}{4t}.$$

Our estimation algorithm, whose performance is characterized by Theorem 1, proceeds via the *method of moments*. Given $X_1, \ldots, X_n$ with $X_i \sim \text{Binomial}(t, p_i)$, and sufficiently large $n$, we can

obtain accurate estimates of the first $t$ moments of the distribution/histogram $P$ defined by the $p_i$'s. Accurate estimates of the first $t$ moments can then be leveraged to recover an estimate of $P$ that is accurate to error $\frac{1}{t}$ plus a factor that depends (exponentially on $t$) on the error in the recovered moments.

The intuition for the lower bound, Proposition 1, is that the realizations of Binomial$(t, p_i)$ give *no* information beyond the first $t$ moments. Additionally, there exist distributions $P$ and $Q$ whose first $t$ moments agree exactly, but which differ in their $t + 1^{st}$ moment, and have $\|P - Q\|_W \geq \frac{1}{2t}$. Putting these two pieces together establishes the lower bound.

We also extend our results to the practically relevant multi-parameter analog of the setting described above, where the $i^{th}$ datapoint corresponds to a pair, or $d$-tuple of hidden parameters, $p_{(i,1)}, \ldots, p_{(i,d)}$, and we observe independent random variables $X_{(i,1)}, \ldots, X_{(i,d)}$ with $X_{(i,j)} \sim$ Binomial$(t_{(i,j)}, p_{(i,j)})$. In this setting, the goal is to recover the multivariate set of $d$-tuples $\{p_{(i,1)}, \ldots, p_{(i,d)}\}$, again in an earth mover's sense. This setting corresponds to recovering an approximation of an underlying joint distribution over these $d$-tuples of parameters.

To give one concrete motivation for this problem, consider a hypothetical setting where we have $n$ genotypes (sets of genetic features), with $t_i$ people of the $i$th genotype. Let $X_{(i,1)}$ denote the number of people with the $i$th genotype who exhibit disease 1, and $X_{(i,2)}$ denote the number of people with genotype $i$ who exhibit disease 2. The interpretation of the hidden parameters $p_{i,1}$ and $p_{i,2}$ are the respective probabilities of people with the $i^{th}$ genotype of developing each of the two diseases. Our results imply that provided $n$ is large, one can accurately recover an approximation to the underlying set or two-dimensional joint distribution of $\{(p_{i,1}, p_{i,2})\}$ pairs, even in settings where there are too few people of each genotype to accurately determine which of the genotypes are responsible for elevated disease risk. Recovering this set of pairs would allow one to infer whether there are common genetic drivers of the two diseases—even in the regime where there is insufficient data to resolve *which* genotypes are the common drivers.

Our multivariate analog of Theorem 1 is also formulated in terms of multivariate analog of earth mover's distance (see Definition 1 for a formal definition):

**Theorem 2.** *Let $\{p_{i,j}\}$ denote a set of $n$ $d$-tuples of hidden parameters in $[0,1]^d$, with $i \in \{1, \ldots, n\}$ and $j \in \{1, \ldots, d\}$, and suppose we observe random variables $X_{i,j}$, with $X_{i,j} \sim$ Binomial$(t, p_{i,j})$. There is an algorithm that produces a distribution $Q$ supported on $[0,1]^d$, such that with probability at least $1 - \delta$ over the randomness of the $X_{i,j}s$,*

$$\|P - Q\|_W \leq \frac{C_1}{t} + C_2 \sum_{|\alpha|=1}^{t} \frac{d(2t)^{d+1} 2^t}{3^{|\alpha|}} \sqrt{\ln(\frac{1}{\delta}) \frac{1}{n}} \leq \frac{C_1}{t} + O_{\delta,t,d}(\frac{1}{\sqrt{n}}),$$

*for absolute constants $C_1, C_2$, where $\alpha$ is a $d$-dimensional multi-index consisting of all $d$-tuples of nonnegative integers summing to at most $t$, $P$ denotes the distribution that places mass $\frac{1}{n}$ at value $p_i = (p_{i,1}, \ldots, p_{i,d}) \in [0,1]^d$, and $\|\cdot\|_W$ denotes the $d$-dimensional Wasserstein distance between $P$ and $Q$.*

## 1.1 Related Work

The seminal paper of Charles Stein [14] was one of the earliest papers to identify the surprising possibility of leveraging the availability of independent data reflecting a large number of parameters of interest, to partially compensate for having little information about each parameter. The specific setting examined considered the problem of estimating a list of unknown means, $\mu_1, \ldots, \mu_n$ given access to $n$ independent Gaussian random variables, $X_1, \ldots, X_n$, with $X_i \sim \mathcal{N}(\mu_i, 1)$. Stein showed that, perhaps surprisingly, that there is an estimator for the list of parameters $\mu_1, \ldots, \mu_n$ that has smaller expected squared error than the naive unbiased empirical estimates of $\hat{\mu}_i = X_i$. This improved estimator "shrinks" the empirical estimates towards the average of the $X_i$'s. In our setting, the process of recovering the set/histogram of unknown $p_i$'s and then leveraging this recovered set as a prior to correct the empirical estimates of each $p_i$ can be viewed as an analog of Stein's "shrinkage", and will have the property that the empirical estimates are shifted (in a non-linear fashion) towards the average of the $p_i$'s.

More closely related to the problem considered in this paper is the work on recovering an approximation to the unlabeled *set* of probabilities of domain elements, given independent draws from a

distribution of large discrete support (see e.g. [11, 2, 15, 16, 1]). Instead of learning the distribution, these works considered the alternate goal of simply returning an approximation to the multiset of probabilities with which the domain elements arise but without specifying which element occurs with which probability. Such a multiset can be used to estimate useful properties of the distribution that do not depend on the labels of the domain of the distribution, such as the entropy or support size of the distribution, or the number of elements likely to be observed in a new, larger sample [12, 17]. The benefit of pursuing this weaker goal of returning the unlabeled multiset is that it can be learned to significantly higher accuracy for a given sample size—essentially as accurate as the empirical distribution of a sample that is a logarithmic factor larger [15, 17].

Building on the above work, the recent work [18] considered the problem of recovering the "frequency spectrum" of rare genetic variants. This problem is similar to the problem we consider, but focuses on a rather different regime. Specifically, the model considered posits that each location $i = 1, \ldots, n$ in the genome has some probability $p_i$ of being mutated in a given individual. Given the sequences of $t$ individuals, the goal is to recover the set of $p_i$'s. The work [18] focused on the regime in which many of the $p_i$'s are significantly less than $\frac{1}{nt}$, and hence correspond to mutations that have never been observed; one conclusion of that work was that one can accurately estimate the number of such rare mutations that would be discovered in larger sequencing cohorts. Our work, in contrast, focuses on the regime where the $p_i$'s are constant, and do not scale as a function of $n$, and the results are incomparable.

Also related to the current work are the works [9, 10] on *testing* whether certain properties of collections of distributions hold. The results of these works show that specific properties, such as whether most of the distributions are identical versus have significant variation, can be decided based on a sample size that is significantly sublinear in the number of distributions.

Finally, the papers [5, 6] consider the related by more difficult setting of learning "Poisson Binomials," namely a sum of independent non-identical Bernoulli random variables, given access to samples. In contrast to our work, in the setting they consider, each "sample" consists of only the sum of these $n$ random variables, rather than observing the outcome of each random variable.

## 1.2 Organization of paper

In Section 2 we describe the two components of our algorithm for recovering the population of Bernoulli parameters: obtaining accurate estimates of the low-order moments (Section 2.1), and leveraging those moments to recover the set of parameters (Section 2.3). The complete algorithm is presented in Section 2.2, and a discussion of the multi-dimensional extension to which Theorem 2 applies is described in Section 2.4. In Section 3 we validate the empirical performance of our approach on synthetic data, as well as illustrate its potential applications to several real-world settings.

## 2 Learning a population of binomial parameters

Our approach to recovering the underlying distribution or set of $p_i$'s proceeds via the method of moments. In the following section we show that, given $\geq t$ samples from each Bernoulli distribution, we can accurately estimate each of the first $t$ moments. In Section 2.3 we explain how these first $t$ moments can then be leveraged to recover the set of $p_i$'s, to earth mover's distance $O(1/t)$.

### 2.1 Moment estimation

Our method-of-moments approach proceeds by estimating the first $t$ moments of $P$, namely $\frac{1}{n}\sum_{i=1}^{n} p_i^k$, for each integer $k$ between 1 and $t$. The estimator we describe is unbiased, and also applies in the setting of Corollary 1 where each $p_i$ is drawn i.i.d. from a distribution $P_{pop}$. In this case, we will obtain an unbiased estimator for $\mathbb{E}_{p \leftarrow P_{pop}}[p^k]$. We limit ourselves to estimating the first $t$ moments because, as show in the proof of the lower bound, Proposition 1, the distribution of the $X_i$'s are determined by the first $t$ moments, and hence no additional information can be gleaned regarding the higher moments.

For $1 \leq k \leq t$, our estimate for the $k^{th}$ moment is $\beta_k = \frac{1}{n}\sum_{i=1}^{n} \frac{\binom{X_i}{k}}{\binom{t}{k}}$. The motivation for this unbiased estimator is the following: Note that given any $k$ i.i.d. samples of a variable distributed

according to Bernoulli($p_i$), an unbiased estimator for $p_i^k$ is their product, namely the estimator which is 1 if all the tosses come up heads, and otherwise is 0. Thus, if we average over all $\binom{t}{k}$ subsets of size $k$, and then average over the population, we still derive an unbiased estimator.

**Lemma 1.** *Given* $\{p_1, \ldots, p_n\}$, *let* $X_i$ *denote the random variable distributed according to Binomial*$(t, p_i)$. *For* $k \in \{1, \ldots, t\}$, *let* $\alpha_k = \frac{1}{n} \sum_{i=1}^n p_i^k$ *denote the* $k^{th}$ *true moment, and* $\beta_k = \frac{1}{n} \sum_{i=1}^n \frac{\binom{X_i}{k}}{\binom{t}{k}}$ *denote our estimate of the kth moment. Then* $\mathbb{E}[\beta_k] = \alpha_k$, *and* $\Pr(|\beta_k - \alpha_k| \geq \epsilon) \leq 2e^{-\frac{1}{3} n \epsilon^2}$.

Given the above lemma, we obtain the fact that, with probability at least $1 - \delta$, the events $|\alpha_k - \beta_k| \leq \sqrt{\ln(\frac{2t}{\delta}) \frac{3}{n}}$ simultaneously occur for all $k \in \{1, \ldots, t\}$.

## 2.2 Distribution recovery from moment estimates

Given the estimates of the moments of the distribution $P$, as described above, our algorithm will recover a distribution, $Q$, whose moments are close to the estimated moments. We propose two algorithms, whose distribution recoveries are via the standard linear programming or quadratic programming approaches which will recover a distribution $Q$ supported on some (sufficiently fine) $\epsilon$-net of $[0, 1]$: the variables of the linear (or quadratic) program correspond to the amount of probability mass that $Q$ assigns to each element of the $\epsilon$-net, the constraints correspond to ensuring that the amount of mass at each element is nonnegative and that the total amount of mass is 1, and the objective function will correspond to the (possibly weighted) sum of the discrepancies between the estimated moments, and the moments of the distribution represented by $Q$.

To see why it suffices to solve this program over an $\epsilon$-net of the unit interval, note that any distribution over $[0, 1]$ can be rounded so as to be supported on an $\epsilon$-net, while changing the distribution by at most $\frac{\epsilon}{2}$ in Wasserstein distance. Additionally, such a rounding alters each moment by at most $O(\epsilon)$, because the rounding alters the individual contributions of point masses to the $k^{th}$ moment by only $O(\epsilon^k) < O(\epsilon)$. As our goal is to recover a distribution with distance $O(1/t)$, it suffices to choose and $\epsilon$-net with $\epsilon \ll 1/t$ so that the additional error due to this discretization is negligible. As this distribution recovery program has $O(1/\epsilon)$ variables and $O(t)$ constraints, both of which are independent of $n$, this program can be solved extremely efficiently both in theory and in practice.

We formally describe this algorithm below, which takes as input $X_1, \ldots, X_n$, binomial parameter $t$, an integer $m$ corresponding to the size of the $\epsilon$-net, and a weight vector $w$.

---

**Algorithms 1 and 2: Distribution Recovery with Linear / Quadratic Objectives**
**Input:** Integers $X_1, \ldots, X_n$, integers $t$ and $m$, and weight vector $w \in \mathbb{R}^t$.
**Output:** Vector $q = (q_0, \ldots, q_m)$ of length $m + 1$, representing a distribution with probability mass $q_i$ at value $\frac{i}{m}$.

- For each $k \in \{1, \ldots, t\}$, compute $\beta_k = \frac{1}{n} \sum \frac{\binom{X_i}{k}}{\binom{t}{k}}$.

- (Algorithm 1) Solve the linear program over variables $q_0, \ldots, q_m$:

$$\text{minimize: } \sum_{k=1}^t |\hat{\beta}_k - \beta_k| w_k, \text{ where } \hat{\beta}_k = \sum_{i=0}^m q_i (\frac{i}{m})^k,$$

$$\text{subject to: } \sum_i q_i = 1, \text{ and for all } i, q_i \geq 0.$$

- (Algorithm 2) Solve the quadratic program over variables $q_0, \ldots, q_m$:

$$\text{minimize: } \sum_{k=1}^t (\hat{\beta}_k - \beta_k)^2 w_k^2, \text{ where } \hat{\beta}_k = \sum_{i=0}^m q_i (\frac{i}{m})^k,$$

$$\text{subject to: } \sum_i q_i = 1, \text{ and for ll } i, q_i \geq 0.$$

---

### 2.2.1 Practical considerations

Our theoretical results, Theorem 1 and Corollary 1, apply to the setting where the weight vector, $w$ in the above linear program objective function has $w_k = 1$ for all $k$. It makes intuitive sense to penalize the discrepancy in the $k$th moment inversely proportionally to the empirically estimated standard deviation of the $k^{th}$ moment estimate, and our empirical results are based on such a weighted objective.

Additionally, in some settings we observed an empirical improvement in the robustness and quality of the recovered distribution if one averages the results of running Algorithm 1 or 2 on several random subsamples of the data. In our empirical section, Section 3, we refer to this as a *bootstrapped* version of our algorithm.

## 2.3 Close moments imply close distributions

In this section we complete the high-level proof that Algorithm 1 accurately recovers $P$, the distribution corresponding to the set of $p_i$'s, establishing Theorem 1 and Corollary 1. The guarantees of Lemma 1 ensure that, with high probability, the estimated moments will be close to the true moments. Together with the observation that discretizing $P$ to be supported on an $\epsilon$-net of $[0, 1]$ alters the moments by $O(\epsilon)$, it follows that there is a solution to the linear program in the second step of Algorithm 1 corresponding to a distribution whose moments are close to the true moments of $P$, and hence with high probability Algorithm 1 will return such a distribution.

To conclude the proof, all that remains is to show that, provided the distribution $Q$ returned by Algorithm 1 has similar first $t$ moments to the true distribution, $P$, then $P$ and $Q$ will be close in Wasserstein (earth mover's) distance. We begin by formally defining the Wasserstein (earth mover's) distance between two distributions $P$ and $Q$:

**Definition 1.** *The Wasserstein, or earth mover's, distance between distributions $P, Q$, is $||P - Q||_W := \inf\limits_{\gamma \in \Gamma(P,Q)} \int_{[0,1]^{2d}} d(x, y) d\gamma(x, y)$, where $\Gamma(P, Q)$ is the set of all couplings on $P$ and $Q$, namely a distribution whose marginals agree with the distributions. The equivalent dual definition is $||P - Q||_W := \sup\limits_{g \in Lip(1)} \int_g(x) d(P - Q)(x)$ where the supremum is taken over Lipschitz functions, g.*

As its name implies, this distance metric can be thought of as the cost of the optimal scheme of "moving" the probability mass from $P$ to create $Q$, where the cost per unit mass of moving from probability $x$ and $y$ is $|x - y|$. Distributions over $\mathbb{R}$, it is not hard to see that this distance is exactly the $\ell_1$ distance between the associated cumulative distribution functions.

The following slightly stronger version of Proposition 1 in [7] bounds the Wasserstein distance between any pair of distributions in terms of the discrepancies in their low-order moments:

**Theorem 3.** *For two distributions $P$ and $Q$ supported on [0, 1] whose first $t$ moments are $\boldsymbol{\alpha}$ and $\boldsymbol{\beta}$ respectively, the Wasserstein distance $||P - Q||_W$ is bounded by $\frac{\pi}{t} + 3^t \sum_{k=1}^{t} |\alpha_k - \beta_k|$.*

The formal proof of this theorem is provided in the Appendix A, and we conclude this section with an intuitive sketch of this proof. For simplicity, first consider the setting where the two distributions $P, Q$ have the *exact* same first $t$ moments. This immediately implies that for any polynomial $f$ of degree at most $t$, the expectation of $f$ with respect to $P$ is equal to the expectation of $f$ with respect to $Q$. Namely, $\int f(x)(P(x) - Q(x))dx = 0$. Leveraging the definition of Wasserstein distance $||P - Q||_W = \sum_{g \in Lip} \int g(x)(P(x) - Q(x))dx$, the theorem now follows from the standard fact that, for any Lipschitz function $g$, there exists a degree $t$ polynomial $f_g$ that approximates it to within $\ell_\infty$ distance $O(1/t)$ on the interval $[0, 1]$.

If there is nonzero discrepancy between the first $t$ moments of $P$ and $Q$, the above proof continues to hold, with an additional error term of $\sum_{k=1}^{t} c_k(\alpha_k - \beta_k)$, where $c_k$ is the coefficient of the degree $k$ term in the polynomial approximation $f_g$. Leveraging the fact that any Lipschitz function $g$ can be approximated to $\ell_\infty$ distance $O(1/t)$ on the unit interval using a polynomial with coefficients bounded by $3^t$, we obtain Theorem 3.

## 2.4 Extension: multivariate distribution estimation

We also consider the natural multivariate extension of the the problem of recovering a population of Bernoulli parameters. Suppose, for example, that every member $i$ of a population of size $n$ has two associated binomial parameters $p_{(i,1)}, p_{(i,2)}$, as in Theorem 2. One could estimate the marginal distribution of the $p_{(i,1)}$ and $p_{(i,2)}$ separately using Algorithm 1, but it is natural to also want to estimate the joint distribution up to small Wasserstein distance in the 2-d sense. Similarly, one can consider the analogous $d$-dimensional distribution recovery question.

The natural idea underlying our extension to this setting is to include estimates of the multivariate moments represented by multi-indices $\alpha$ with $|\alpha| \leq t$. For example, in a 2-d setting, the moments for members $i$ of the population would look like $\mathbb{E}_{p_i \sim P}[p_{(i,1)}^a p_{(i,2)}^b]$. Again, it remains to bound how close an interpolating polynomial can get to any $d$-dimensional Lipschitz function, and bound the size of the coefficients of such a polynomial. To this end, we use the following theorem from [3]:

**Lemma 2.** *Given any Lipschitz function $f$ supported on $[0,1]^d$, there is a degree $s$ polynomial $p(x)$ such that*

$$\sup_{x \in [0,1]^d} |p(x) - f(x)| \leq \frac{C_d}{t},$$

*where $C_d$ is a constant that depends on $d$.*

In Appendix D, we prove the following bound on the magnitude of the coefficients of the interpolating polynomial: $|c_\alpha| \leq \frac{(2t)^d 2^t}{3^{|\alpha|}}$, where $c_\alpha$ is the coefficient of the $\alpha$ multinomial term. Together with the concentration bound of the $\alpha^{th}$ moment of the distribution, we obtain Theorem 2, the multivariate analog of Theorem 1.

## 3 Empirical performance

### 3.1 Recovering distributions with known ground truth

We begin by demonstrating the effectiveness of our algorithm on several synthetic datasets. We considered three different choices for an underlying distribution $P_{pop}$ over $[0,1]$, then drew $n$ independent samples $p_1, \ldots, p_n \leftarrow P_{pop}$. For a parameter $t$, for each $i \in \{1, \ldots, n\}$, we then drew $X_i \leftarrow Binomial(t, p_i)$, and ran our population estimation algorithm on the set $X_1, \ldots, X_n$, and measured the extent to which we recovered the distribution $P_{pop}$. In all settings, $n$ was sufficiently large that there was little difference between the histogram corresponding to the set $\{p_1, \ldots, p_n\}$ and the distribution $P_{pop}$. Figure 1 depicts the error of the recovered distribution as $t$ takes on all even values from 2 to 14, for three choices of $P_{pop}$: the "3-spike" distribution with equal mass at the values $1/4, 1/2$, and $3/4$, a Normal distribution truncated to be supported on $[0,1]$, and the uniform distribution over $[0,1]$.

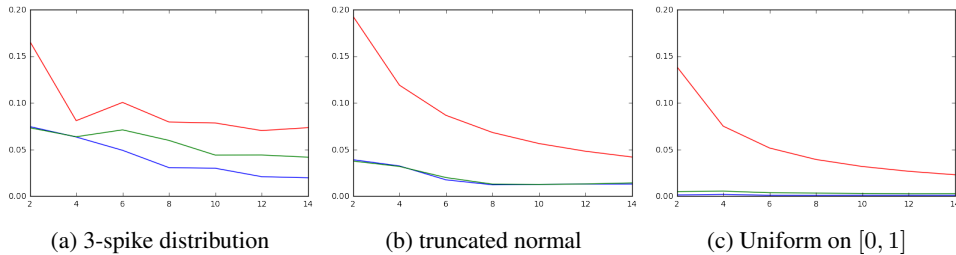

(a) 3-spike distribution      (b) truncated normal      (c) Uniform on $[0,1]$

Figure 1: Earth mover's distance (EMD) between the true underlying distribution $P_{pop}$ and the distribution recovered by Algorithm 2 for three choices of $P_{pop}$: (a) the distribution consisting of equally weighted point masses at locations $\frac{1}{4}, \frac{1}{2}, \frac{3}{4}$; (b) the normal distribution with mean 0.5 and standard deviation 0.15, truncated to be supported on $[0,1]$; and (c) the uniform distribution over $[0,1]$. For each underlying distributions, we plot the EMD (median over 20 trials) between $P_{pop}$ and the distribution recovered with Algorithm 2 as $t$, the number of samples from each of the $n$ Bernoulli random variables, takes on all even values from 2 to 14. These results are given for $n = 10,000$ (green) and $n = 100,000$ (blue). For comparison, the distance between $P_{pop}$ and the histogram of the empirical probabilities for $n = 100,000$ is also shown (red).

Figure 2 shows representative plots of the CDFs of the recovered histograms and empirical histograms for each of the three choices of $P_{pop}$ considered above.

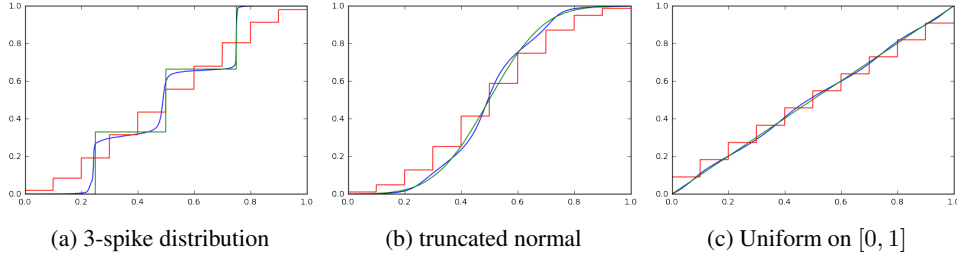

(a) 3-spike distribution    (b) truncated normal    (c) Uniform on $[0, 1]$

Figure 2: CDFs of the true distribution $P$ (green), the histogram recovered by Algorithm 2 (blue) for $P$, and the empirical histogram (red) corresponding to $t = 10$ samples and $n = 100,000$. Note that the empirical distribution is only supported on multiples of $\frac{1}{10}$.

We also considered recovering the distribution of probabilities that different flights are delayed (i.e. each flight—for example *Delta Airlines 123*—corresponds to a parameter $p \in [0, 1]$ representing the probability that flight is delayed on a given day. Our algorithm was able to recover this non-parametric distribution of flight delay parameters extremely well based on few ($\leq 10$) data points per flight. In this setting, we had access to a dataset with $> 50$ datapoints per flight, and hence could compare the recovered distribution to a close approximation of the ground truth distribution. These results are included in the appendix.

## 3.2 Distribution of offspring sex ratios

One of the motivating questions for this work was the following naive sounding question: do all members of a given species have the same propensity of giving birth to a male vs female child, or is there significant variation in this probability across individuals? For a population of $n$ individuals, letting $p_i$ represent the probability that a future child of the $i$th individual is male, this questions is precisely the question of characterizing the histogram or set of the $p_i$'s. This question of the uniformity of the $p_i$'s has been debated both by the popular science community (e.g. the recent BBC article "Why Billionaires Have More Sons"), and more seriously by the biology community.

Meiosis ensures that each male produces the same number of spermatozoa carrying the X chromosome as carrying the Y chromosome. Nevertheless, some studies have suggested that the difference in the amounts of genetic material in these chromosomes result in (slight) morphological differences between the corresponding spermatozoa, which in turn result in differences in their motility (speed of movement), etc. (see e.g. [4, 13]). Such studies have led to a chorus of speculation that the relative timing of ovulation and intercourse correlates with the sex of offspring.

While it is problematic to tackle this problem in humans (for a number of reasons, including sex-selective abortions), we instead consider this question for dogs. Letting $p_i$ denote the probability that each puppy in the $i$th litter is male, we could hope to recover the distribution of the $p_i$'s. If this sex-ratio varies significantly according to the specific parents involved, or according to the relative timing of ovulation and intercourse, then such variation would be evident in the $p_i$'s. Conveniently, a typical dog litter consists of 4-8 puppies, allowing our approach to recover this distribution based on accurate estimates of these first moments.

Based on a dataset of $n \approx 8,000$ litters, compiled by the Norwegian Kennel Club, we produced estimates of the first 10 moments of the distribution of $p_i$'s by considering only litters consisting of at least 10 puppies. Our algorithm suggests that the distribution of the $p_i$'s is indistinguishable from a spike at $\frac{1}{2}$, given the size of the dataset. Indeed, this conclusion is evident based even on the estimates of the first two moments: $\frac{1}{n}\sum_i p_i \approx 0.497$ and $\frac{1}{n}\sum_i p_i^2 \approx 0.249$, since among distribution over $[0, 1]$ with expectation $1/2$, the distribution consisting of a point mass at $1/2$ has minimal variance, equal to $0.25$, and these two moments robustly characterize this distribution. (For example, any distribution supported on $[0, 1]$ with mean $1/2$ and for which $> 10\%$ of the mass lies outside the range $(0.45, 0.55)$, must have second moment at least $0.2505$, though reliably resolving such small variation would require a slightly large dataset.)

### 3.3 Political tendencies on a county level

We performed a case study on the political leanings of counties. We assumed the following model: Each of the $n = 3116$ counties in the US have an intrinsic "political-leaning" parameter $p_i$ denoting their likelihood of voting Republican in a given election. We observe $t = 8$ independent samples of each parameter, corresponding to whether each county went Democratic or Republican during the 8 presidential elections from 1976 to 2004.

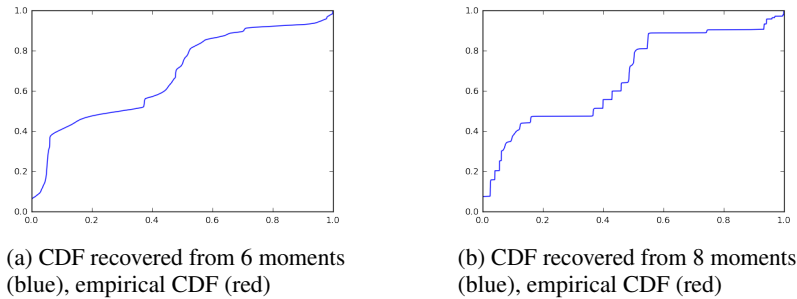

(a) CDF recovered from 6 moments (blue), empirical CDF (red)

(b) CDF recovered from 8 moments (blue), empirical CDF (red)

Figure 3: Output of bootstrapping Algorithm 2 on political data for $n =$3,116 counties over $t = 8$ elections.

### 3.4 Game-to-game shooting of NBA players

We performed a case study on the scoring probabilities of two NBA players. One can think of this experiment as asking whether NBA players, game-to-game, have differences in their intrinsic ability to score field goals (in the sports analytics world, this is the idea of "hot / cold" shooting nights). The model for each player is as follows: for the $i$th basketball game there is some parameter $p_i$ representing the player's latent shooting percentage for that game, perhaps varying according to the opposing team's defensive strategy. The empirical shooting percentage of a player varies significantly from game-to-game—recovering the underlying distribution or histogram of the $p_i$'s allows one to directly estimate the consistency of a player. Additionally, such a distribution could be used as a prior for making decisions during games. For example, conditioned on the performance during the first half of a game, one could update the expected fraction of subsequent shots that are successful.

The dataset used was the per-game 3 point shooting percentage of players, with sufficient statistics of "3 pointers made" and "3 pointers attempted" for each game. To generate estimates of the $k^{th}$ moment, we considered games where at least $k$ 3 pointers were attempted. The players chosen were Stephen Curry of the Golden State Warriors (who is considered a very consistent shooter) and Danny Green of the San Antonio Spurs (whose nickname "Icy Hot" gives a good idea of his suspected consistency).

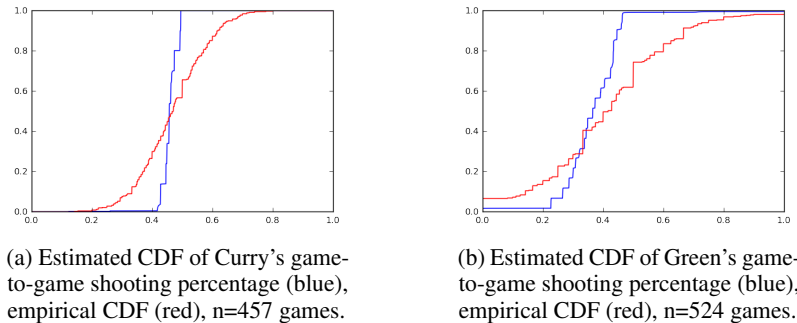

(a) Estimated CDF of Curry's game-to-game shooting percentage (blue), empirical CDF (red), n=457 games.

(b) Estimated CDF of Green's game-to-game shooting percentage (blue), empirical CDF (red), n=524 games.

Figure 4: Estimates produced by bootstrapped version of Algorithm 2 on NBA dataset, 8 moments included

## Acknowledgments

We thank Kaja Borge and Ane Nødtvedt for sharing an anonymized dataset on sex composition of dog litters, based on data collected by the Norwegian Kennel Club. This research was supported by NSF CAREER Award CCF-1351108, ONR Award N00014-17-1-2562, NSF Graduate Fellowship DGE-1656518, and a Google Faculty Fellowship.

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
