[Supplementary Material]

# A Proof of Theorem 3, the Wasserstein distance bound

**Theorem 3** *For two distributions $P$ and $Q$ supported on [0, 1] whose first $t$ moments are $\boldsymbol{\alpha}$ and $\boldsymbol{\beta}$ respectively, the Wasserstein distance $||P - Q||_W$ is bounded by $\frac{\pi}{t} + 3^t \sum_{k=1}^t |\alpha_k - \beta_k|$.*

*Proof.* The natural approach to bounding the Wasserstein distance,

$$\sup_{f \in Lip_1} \int f(x)\, (P(x) - Q(x))\, dx,$$

is to argue that for any Lipschitz function, $f$, there is a polynomial $P_f$ of degree at most $k$ that closely approximates $f$. To see this,

$$\int_0^1 f(x)(P(x) - Q(x))dx$$

$$\leq \int_0^1 |p_f(x) - f(x)|(P(x) - Q(x))dx + \int_0^1 p_f(x)(P(x) - Q(x))dx$$

$$\leq 2||f - p_f||_\infty + \sum_{k=1}^t c_k(\alpha_k - \beta_k),$$

where $c_k$ be the coefficient of the degree-$k$ term of polynomial $p_f$. Hence all that remains is to argue that there is a good degree $k$ polynomial approximation of any Lipschitz function $f$.

For convenience of the analysis, we generalize the domain of $f$ from $[0, 1]$ to $[-1, 1]$ by letting $f(-x) = f(x)$. We further define function $\phi(\theta) = f(\cos(\theta))$ which also has Lipschitz constant 1 since the cosine function has Lipschitz constant 1. Now we are ready to apply Theorem 4.2.1 of [8] to $\phi(\theta)$, which states that for any periodic-$2\pi$ function with Lipschitz constant 1 can be approximate by a degree $t$ trigonometric polynomials with $l_\infty$ approximation error $\frac{K_1}{t} = \frac{\pi}{2t}$ where $K_1$ is Favard constant which is equal to $\frac{\pi}{2}$. Let $U_n(\theta)$ be the degree $t$ trigonometric polynomials that achieves the stated approximation error. WLOG, by Proposition 2.1.6 of [8], we may assume $U_n(\theta)$ is even. The algebraic polynomial to approximate $f(x)$ can be defined as $p_f(x) = U_t(\arccos(x))$ which again has degree $t$. Hence we have shown that $||f - p_f||_\infty \leq \frac{\pi}{2t}$ and what remains is to bound the magnitude of $c_k$.

The plan is to first obtain sharp bound of the coefficients of the trigonometric polynomials $U_t(\theta)$ explicitly, after which $c_k$ can be bounded by being expressed in terms of these coefficients. Notice that the coefficient of term $\cos(k\theta)$ in $U_t(\theta)$, denoted as $u_k$, is $a_k \lambda_k^t$ by Formula 1.1 in Chapter 4 of [8] where $a_k = \frac{1}{\pi} \int_0^{2\pi} \phi(\theta) \cos(k\theta) d\theta$ and $\lambda_k^t = \frac{k\pi}{2(t+1)} \frac{1}{\tan(\frac{k\pi}{2(t+1)})}$ by Formula 1.42 in Chapter 4 of [8].

Given that $\tan(x) \geq x$ for $0 \leq x \leq \frac{\pi}{2}$ and $\frac{k\pi}{2(t+1)} < \frac{\pi}{2}$, we have $\frac{1}{\tan(\frac{k\pi}{2(t+1)})} \leq \frac{2(t+1)}{k\pi}$ and hence $0 \leq \lambda_k^t \leq 1$. In order to bound $a_k$, notice that WLOG, we may assume $||f||_\infty \leq 1/2$ and $||\phi||_\infty \leq 1/2$ since $f$ is Lipschitz-1. Hence $|a_k| = \frac{1}{\pi}|\int_0^{2\pi} \phi(\theta) \cos(k\theta) d\theta| \leq \frac{1}{2\pi} \int_0^{2\pi} |\cos(k\theta)| d\theta \leq 1$. We have shown that for all $k$, $u_k$ is at most 1.

The algebraic polynomial $U_t(\arccos(x))$ can be expressed as $\sum_{k=1}^t u_k T_k(x)$ where $T_k(x)$ is Chebyshev polynomials of the first kind. Note the recurrence relation for Chebyshev polynomials given by $T_{n+1}(x) = 2x T_n(x) - T_{n-1}(x), T_0(x) = 1, T_1(x) = x$, for the $i$th polynomial, we can loosely bound the magnitude of any of its coefficients by $3^{i-1}$. Since $|u_i| < 1$ for all $i$, the magnitude of coefficient $c_k$ can be upper bounded by $\sum_{i=1}^t 3^{i-1} \leq 3^t$. Thus, we have shown that:

$$\int_0^1 f(x)(P(x) - Q(x))dx \leq \frac{\pi}{t} + 3^t \sum_{k=1}^t |\alpha_k - \beta_k|.$$

$\square$

# B Proof of Theorem 1

In this section, we prove the main theorem of our paper, Theorem 1, which establishes guarantees of the estimation accuracy of our algorithm. Before proving our main theorem, we first prove Lemma 1, the properties of our moment estimators:

**Lemma 1** *Given* $\{p_1, \ldots, p_n\}$, *let* $X_i$ *denote the random variable distributed according to* $Binomial(t, p_i)$. *For* $k \in \{1, \ldots, t\}$, *let* $\alpha_k = \frac{1}{n} \sum_{i=1}^{n} p_i^k$ *denote the* $k^{th}$ *true moment, and* $\beta_k = \frac{1}{n} \sum_{i=1}^{n} \frac{\binom{X_i}{k}}{\binom{t}{k}}$ *denote our estimate of the kth moment. Then*

$$\mathbb{E}[\beta_k] = \alpha_k, \text{ and } \Pr(|\beta_k - \alpha_k| \geq \epsilon) \leq 2e^{-\frac{1}{3}n\epsilon^2}.$$

*Proof.* First we show that for each $i$ we have $\mathbb{E}[\binom{X_i}{k}] = p_i^k \binom{t}{k}$, then the claim $\mathbb{E}[\beta_k] = \alpha_k$ holds trivially due to the additivity of expectation. Notice that the numerator counts the number of subsets of size $k$ that are all 1, and the denominator is the number of subsets of size $k$. The probability that a certain subset of size $k$ is all 1 is exactly $p_i^k$. Hence the claim about the expectation holds.

By Bernstein's Inequality, when $\epsilon \leq 1$, $\Pr(|\beta_k - \alpha_k| \geq \epsilon) \leq 2e^{-\frac{3}{8}n\epsilon^2} \leq 2e^{-\frac{1}{3}n\epsilon^2}$ holds. We have proved the claim about concentration. $\square$

We are now ready to prove Theorem 1. For convenience, we restate the theorem:

**Theorem 1** *Consider a set of $n$ probabilities, $p_1, \ldots, p_n$ with $p_i \in [0, 1]$, and suppose we observe the outcome of $t$ independent flips of each coin, namely $X_1, \ldots, X_n$, with $X_i \sim Binomial(t, p_i)$. There is an algorithm that produces a distribution $Q$ supported on $[0, 1]$, such that with probability at least $1 - \delta$ over the randomness of $X_1, \ldots, X_n$,*

$$\|P - Q\|_W \leq \frac{\pi}{t} + 3^t \sum_{i=1}^{t} \sqrt{\ln(\frac{2t}{\delta})\frac{3}{n}} \leq \frac{\pi}{t} + O_\delta(\frac{3^t t \ln t}{\sqrt{n}}),$$

*where $P$ denotes the distribution that places mass $\frac{1}{n}$ at value $p_i$, and $\|\cdot\|_W$ denotes the Wasserstein distance.*

*Proof.* Given Lemma 1, we obtain the fact that, with probability at least $1 - \delta$, the events $|\alpha_k - \beta_k| \leq \sqrt{\ln(\frac{2t}{\delta})\frac{3}{n}}$ simultaneously occur for all $k \in \{1, \ldots, t\}$. Applying Theorem 3 yields the claimed accuracy guarantee. $\square$

## C   Proof of Proposition 1, the information-theoretic lower bound

In this section, we prove Proposition 1 establishing the tightness of the $\Theta(1/t)$ dependence in our recovery guarantees. For convenience, we restate the proposition:

**Proposition 1** *Let $P_{pop}$ denote a distribution over $[0, 1]$, and for positive integers $t$ and $n$, let $X_1, \ldots, X_n$ denote independent random variables with $X_i$ distributed as $Binomial(t, p_i)$ where $p_i$ is drawn independently according to $P_{pop}$. An estimator $f$ maps $X_1, \ldots, X_n$ to a distribution $f(X_1, \ldots, X_n)$. Then, for every fixed $t$, the following lower bound on the accuracy of any estimator holds for all $n$:*

$$\inf_f \sup_{P_{pop}} \mathbb{E}\left[\|f(X_1, \ldots, X_n) - P_{pop}\|_W\right] > \frac{1}{4t}.$$

Our proof will leverage the following result from [7] which states that there exists a pair of distributions supported on $[0, 1]$ whose first $t$ moments agree, but have Wasserstein distance $> 1/2t$:

**Lemma 3.** *For any $t$, there exists a pair of distributions $D_P, D_Q$ supported on $[0, 1]$ that each consist of $O(t)$ point masses, such that $D_P$ and $D_Q$ have identical first $t$ moments, and $\|D_P - D_Q\|_W > \frac{1}{2t}$*

*Proof of Proposition 1.* Consider the distributions $D_P$ and $D_Q$ whose existence is guaranteed by Lemma 3. Consider the distribution of $X_i$, where $X_i$ is drawn by first drawing $p_i$ according to $D_P$, and then drawing $X_i \leftarrow Binomial(p_i, t)$. Similarly, let $Y_i$ denote the random variable defined by drawing $q_i$ from $D_Q$ and then drawing $Y_i \leftarrow Binomial(q_i, t)$.

We now claim that the distribution of $X_i$ and $Y_i$ are identical, and hence, for every $n$, the joint distribution of $(X_1, \ldots, X_n)$ is identical to that of $(Y_1, \ldots, Y_n)$, and hence they cannot be distinguished.

Indeed, the distributions of $X_i$ and $Y_i$ are given by:

$$\mathbb{P}(X_i = k) = \int_0^1 \binom{t}{k} x^k (1-x)^{t-k} D_P(x) dx$$

$$\mathbb{P}(Y_i = k) = \int_0^1 \binom{t}{k} x^k (1-x)^{t-k} D_Q(x) dx$$

Noting that the integrand is a degree-$t$ polynomial, and that $D_P$ and $D_Q$ have the same first $t$ moments yields the conclusion that these two distributions are identical.

To conclude, note that if we are given $(Z_1, \ldots, Z_n)$ with the promise that, with probability $1/2$, they correspond to $D_P$ and with probability $1/2$ they correspond to $D_Q$, then no algorithm can correctly guess which of these distributions they were drawn from, with probability of success greater than $1/2$, and hence no estimator can achieve an expected error of recovery better than $\frac{1}{2}\|D_P - D_Q|_W > \frac{1}{4t}$, as desired. □

# D    Proof of Theorem 2, multivariate setting

The prove of Theorem 2 will be identical to Theorem 1, except that we will need the following slightly stronger version of Lemma 2:

**Lemma 2** *Given any Lipschitz function $f$ supported on $[0,1]^d$, there is a degree $t$ polynomial $p(x) = \sum_{|\alpha| \leq t} c_\alpha x^\alpha$ where $\alpha$ is multi-index $\{\alpha_1, \alpha_2, \ldots \alpha_d\}$ such that*

$$\sup_{x \in [0,1]^d} |p(x) - f(x)| \leq \frac{C_d}{t}, \tag{1}$$

*and $c_\alpha \leq A_d \frac{(2t)^d 2^t}{3^{|\alpha|}}$.*

*Proof.* This polynomial approximation lemma is basically a restatement of Theorem 1 in [3]. What we need to do is only to give an explicit upper bound of the coefficients.

The high level idea is to first convolve $f$ with a holomorphic bump function $G$ which gives $H = f * G$, then the Maclaurin series of $H$ is a good polynomial approximation of $H$ and also $f$.

By the definition of Maclaurin series, the coefficient $c_\alpha = \frac{\partial^\alpha H(0)}{|\alpha|!}$. Suppose $H$ is holomorphic on an open neighborhood of some polydisk $E_S$ with radius $S$, assuming $\sup_{z \in E_S} |H(z)| \leq M$, by Cauchy's integral formula, we have $|c_\alpha| = |\frac{1}{2\pi i} \oint_{|z|=S} \frac{H(z)}{z^{|\alpha|+1}}| \leq \frac{M}{S^{|\alpha|}}$. By the definition of $R$ in the proof of Theorem 1 in [3], we can set $R = 1$ such that function $f$ is supported on box $B_R$. Let $S = 2R + 1 = 3$ and follow all the parameter settings, by Equation 14 in [3], we have $|c_\alpha| \leq \frac{M}{S^{|\alpha|}} \leq A_d \frac{(2t)^d (t+1) 2^t}{t 3^{|\alpha|}} \leq A_d \frac{(2t)^d 2^t}{3^{|\alpha|}}$, where $A_d$ is a constant that depends on $d$. □

# E    Lateness in flights

We evaluated our recovery algorithm on the distribution of flight delays, based on the 2015 Flight Delays and Cancellations dataset from Kaggle via the Department of Transportation. This dataset contains records consisting of flight identifer (e.g. airline, number, and departure/arrival city), date, lateness of departure, and lateness of arrival. For each of $n = 25,156$ different flights—where a "flight" is defined via the airline and flight number—we let the corresponding binomial parameter $p$ correspond to the probability that flight departs at least 15 minutes late. Each of the $n = 25,156$ flights considered had at least 50 records in the dataset, and we took the empirical distribution of the lateness parameters of these flights as our ground truth distribution, $P_{pop}$.

In our experiment, for each of the $n$ flights, we subsampled $t$ of the corresponding records uniformly at random, and let $X_i$ represent the number of these $t$ sampled instances that departed late. Then we

ran Algorithm 2 on $X_1, \ldots, X_n$ to produce an estimate $Q$ of $P_{pop}$. Figure 5 depicts the results of a typical run of the algorithm corresponding to sample sizes of $t = 6$ and $t = 10$. In the plots, the red line is the CDF of the empirical estimator, the blue line is the CDF of the output of Algorithm 2, and the green line is the ground truth. Based on 5 independent runs of the experiments, the average and standard deviation of the EMD error of the recovered distribution was $\mu = 0.023$ and $std = 0.0051$ ($t = 6$) and $\mu = 0.014$ and $std = 0.0027$ ($t = 10$). For comparison, the average EMD error of the empirical estimates of the distribution were $0.069$ ($t = 6$) and $0.044$ ($t = 10$).

The estimates were very robust to repeated runs of the experiment, producing CDFs that matched the ground truth extremely closely for all settings of $t \in \{4, 5, \ldots, 10\}$.

(a) $t = 6$ samples.          (b) $t = 10$ samples.

Figure 5: Recovering the distribution of probabilities of flight departure delay. The blue line depicts the distribution recovered by bootstrapping Algorithm 2 on $t = 6$ samples of each flight (left plot) and $t = 10$ samples of each flight (right plot). For comparison, the ground truth distribution (green line) and empirical distribution (red line) are also shown.