[Reviews · NeurIPS 2017]

Reviewer 1



The paper considers a statistical framework where many observations are available but only a few concern each individual parameter of interests. Instead of estimating each parameter individually, following an idea of Stein, the idea is to estimate population parameters", that is to provide an estimate of the empirical distribution of these parameters. Theoretically, an estimation of the empirical distribution based on its first moments is proposed and a control of the risk of this estimator with respect to the Wasserstein distance is proved. Then, an algorithm for practical implementation of this estimator is proposed and illustrates the performances of the estimator on both synthetic and real data-sets. The interest of the statistical problem is shown on political tendency and sports performances at the end of the paper. I think that the statistical problem, the way to analyse it and the moment solution are interesting and I strongly recommand the paper for publication. Nevertheless, I think that many references and related works could be mentionned. For example, it seems that the estimator would provide an interesting relevant prior for an empirical Bayes procedure, the connection with latent variables in mixed effects models should be discussed, as well as the connection of this procedure with hidden" variables model like HMM. Finally, a moment procedure was proposed recently to estimate the unobserved random environment" of a Markov Chain. Comparisons with these related works could improved the paper.

Reviewer 2



This paper establishes an interesting new problem setting, that of estimating the distribution of parameters for a group of binomial observations, gives an algorithm significantly better (in earth-mover distance) than the obvious one, and shows both a bound on its performance and an asymptotically-matching lower bounds. It also gives empirical examples on several datasets, showing the algorithm's practical advantages. I did not closely verify the proofs, but they seem reasonable. Presentation-wise, they are somewhat disorganized: it would be helpful to label the sections e.g. Proof of Theorem 3, and provide a formal statement of the proposition being proved in the noise-free case, etc. Unfortunately, the algorithm does involve solving a fairly large linear program. Explicitly characterizing the computational complexity and accuracy effects of m would be helpful. It would also be much nicer to not need a hard cutoff on the moment estimators s. Perhaps you could weight the loss function by the standard error of their estimators, or something similar? The estimators should be asymptotically normal, and you could e.g. maximize the likelihood under that distribution (though their asymptotic correlations might be difficult to characterize). Though it's an interesting problem setting, it also doesn't seem like one likely to take the world by storm; this paper is probably of primarily theoretical interest. Some interesting potential applications are suggested, however. A question about the proof: in lines 400-401, I think some more detail is needed in the proof of the variance bound. Choosing the t/k independent sets would give the stated bound, but of course there are \binom{t}{k} - t/k additional sets correlated to those t/k. It seems likely that averaging over these extra sets decreases the variance, but this is not immediate: for example, if all of the other sets were identical, the variance would be increased. I think some work is needed here to prove this, though maybe there's an obvious reason that I'm missing. Minor notes: Line 47: should be "Stein's phenomenon." Line 143, "there exists only one unbiased estimator": If you add N(0, 1) noise to any unbiased estimator, it's still unbiased. Or you could weight different samples differently, or any of various other schemes. Your proof in Appendix E (which should be referenced by name in the main text, to make it easier to find...) rather shows that there is a unique unbiased estimator which is a deterministic function of the total number of 1s seen. There is of course probably no reason to use any of those other estimators, and yours is the MVUE, but the statement should be corrected. Lines 320-322: (6) would be clearer if you established t to be scaled by c explicitly. y axis limits for CDF figures (2, 3, 4) should be clamped to [0, 1]. Probably the y lower bound in Figure 1a should be 0 as well.

Reviewer 3



In this paper, authors study the problem of learning a set of Binomial parameters in Wasserstein distance and demonstrate that sorted set of probabilities can be learned at a rate of (1/t) where t is the parameter in the Binomial distribution. The result seems interesting and I had a great time reading it. However, I would like to understand the technical novelty of the paper better. Can the authors please expand on the differences between this paper and one of the cited papers: “Instance optimal learning of discrete distributions”, by Valiant and Valiant? To be more specific: In many of the discrete learning problems, Poisson approximation holds i.e., Binomial (t,p_i) ~ Poisson (tp_i). Hence, the proposed problem is very similar to estimating n Poisson random variables with means tp_1, tp_2, \ldots tp_n. Similarly the problem in the VV paper boils down to estimating many Poisson random variables. Hence, aren't they almost the same problem? Furthermore, in VV’s paper, they use empirical estimates for large Poisson random variables (mean > log n), and use a moment based approach for small Poisson random variables (mean < log n). In this paper, authors use moment based approach for small Poisson random variables (mean < t). In both the cases, the moment based approach yields an improvement of 1/poly(mean) ~1/poly(t) improvement in earthmovers / Wasserstein’s distance. Given the above, can you please elaborate the differences between the two papers? Few other questions to the authors: 1. In Theorem 1: What is the dependence of t in the term O_{delta, t}(1/\sqrt{n}). 2. In Theorem 1: s is not known in advance, it is a parameter to be tuned/ supplied to the algorithm. Hence, how do you obtain min_s in the right hand side of the equation? 3. Why Wasserstein’s distance? 4. Good-Turing approach (McAllester Schapire ‘00) seems to perform well for instance optimality of discrete distributions in KL distance (Orlitsky Suresh ‘15). However, VV showed that Good-Turing does not work well for total variation distance. In your problem, you are looking at Wasserstein’s distance. Can you please comment if you think Good-Turing approaches work well for this distance? Can you given example / compare these approaches at least experimentally? Edits after the rebuttal: Changed to Good paper, accept. > For ANY value of t, there exist dists/sets of p_i's such that the VV and Good-Turing approaches would recover dists that are no better (and might be worse?) than the empirical distribution (i.e. they would get a rate of convergence > O(1/sqrt(t)) rather than the O(1/t) that we get). Can you please add these examples in the paper, at least in the appendix? It would be also nice if you can can compare with these approaches experimentally. > Suppose p_i = 0.5 for all i, t=10, and n is large. The analogous "poissonized" setting consists of taking m<--Poi(5n) draws from the uniform distribution over n items, and the expected number of times the ith item is seen is distributed as Poi(5). Poi(5) has variance 5, though Binomial(10,0.5) has variance 10/4 = 2.5, and in general, for constant p, Poi(t*p) and Binomial(t,p) have constant L1 distance and variances that differ by a factor of (1-p). While I agree that they have a constant L1 distance, I still am not convinced that these problems are 'very different'. Do you believe that your algorithm (or a small variation of it) would not work for the problem where the goal is to estimate n Poisson random variables? If so / no, please comment. If not, at least say something about it in the paper. > [A "bad" construction is the following: consider a distribution of p_i's s.t. the corresponding mixture of binomials approximates a Poisson, in which case VV and Good-Turing will conclude wrongly that all the p_i's are identical.] I do not agree with this comment too. Note that for this argument to work it is not sufficient to approximate each Poisson coordinate with a mixture of binomials coordinate. One has to approximate a Poisson product distribution over n coordinates, with a mixture of binomial random variables with n coordinates. I believe this task gets harder as n-> infty. > It depends on the distribution (and is stated in the theorem), but can be as bad as 2^t. Please mention it in the paper.